# Roles of the Hepatic Endocannabinoid and Apelin Systems in the Pathogenesis of Liver Fibrosis

**DOI:** 10.3390/cells8111311

**Published:** 2019-10-24

**Authors:** Pedro Melgar-Lesmes, Meritxell Perramon, Wladimiro Jiménez

**Affiliations:** 1Biochemistry and Molecular Genetics Service, Hospital Clínic Universitari, IDIBAPS, CIBERehd, Villarroel 170, 08036 Barcelona, Spain; pmelgar@clinic.cat (P.M.-L.); mperramon@clinic.cat (M.P.); 2Department of Biomedicine, University of Barcelona, 08036 Barcelona, Spain; 3Institute for Medical Engineering and Science, Massachusetts Institute of Technology, Cambridge, MA 02139, USA

**Keywords:** endocannabinoids, apelin, liver fibrosis, CB_1_, CB_2_, APJ

## Abstract

Hepatic fibrosis is the consequence of an unresolved wound healing process in response to chronic liver injury and involves multiple cell types and molecular mechanisms. The hepatic endocannabinoid and apelin systems are two signalling pathways with a substantial role in the liver fibrosis pathophysiology—both are upregulated in patients with advanced liver disease. Endogenous cannabinoids are lipid-signalling molecules derived from arachidonic acid involved in the pathogenesis of cardiovascular dysfunction, portal hypertension, liver fibrosis, and other processes associated with hepatic disease through their interactions with the CB_1_ and CB_2_ receptors. Apelin is a peptide that participates in cardiovascular and renal functions, inflammation, angiogenesis, and hepatic fibrosis through its interaction with the APJ receptor. The endocannabinoid and apelin systems are two of the multiple cell-signalling pathways involved in the transformation of quiescent hepatic stellate cells into myofibroblast like cells, the main matrix-producing cells in liver fibrosis. The mechanisms underlying the control of hepatic stellate cell activity are coincident despite the marked dissimilarities between the endocannabinoid and apelin signalling pathways. This review discusses the current understanding of the molecular and cellular mechanisms by which the hepatic endocannabinoid and apelin systems play a significant role in the pathophysiology of liver fibrosis.

## 1. Introduction

Hepatic fibrosis is the excessive accumulation of connective tissue proteins, particularly collagens, in the liver extracellular matrix [1]. This process is the result of chronic liver injury that may be of different aetiologies: viruses, ethanol, toxins, drugs, or cholestasis [1]. Non-alcoholic fatty liver disease (NAFLD) is the most common cause of chronic liver disease worldwide with a prevalence of 20 to 40% in the general population and up to 95% in subjects with obesity and diabetes [2]. NAFLD is characterized by an abnormal accumulation of fatty acids into hepatocytes and includes a wide spectrum of liver diseases, ranging from mild to severe steatosis and non-alcoholic steatohepatitis (NASH) [3]. Both NAFLD and NASH have the potential to evolve into liver fibrosis and cirrhosis. Liver fibrosis involves multiple cell types and molecular mechanisms and processes. Hepatocellular injury unleashes a complex inflammatory response and the release of a plethora of cytokines and growth factors that, on one hand, trigger transformation of quiescent hepatic stellate cells (HSC) into myofibroblast-like cells and, on the other hand, stimulate inflammation and angiogenesis [4]. Chronic inflammation and the long-term injury and regeneration processes perpetuate liver fibrosis and eventually results in distortion of lobular architecture, nodular formation, and cirrhosis [5]. Documented reversibility of advanced liver fibrosis in patients [6,7,8] has encouraged research on anti-fibrotic drugs and novel therapeutic strategies [9,10,11,12,13,14,15], which are effective in experimental models of liver fibrosis, but their utility in humans is still unknown. Most strategies aimed at interfering with liver fibrosis focus on the inhibition of HSC activation, the modulation of inflammatory response, or the reduction of hepatocyte damage by targeting and modulating multiple cell signalling pathways [4]. Two of these signalling pathways, which have significant impact on the pathophysiology of liver fibrosis, are the endocannabinoid and apelin systems. Patients with advanced liver disease show high circulating levels of both endocannabinoids and apelin, and their respective signalling pathways appear to be upregulated in advanced liver disease [16,17,18].

This review focuses on recent advances in elucidating the roles of the endocannabinoid and apelin systems in liver fibrosis and their potential clinical relevance. We emphasize recent findings regarding the cellular and molecular mechanisms by which these endogenous systems are involved in liver fibrosis.

## 2. Overview of the Endogenous Cannabinoid System

Since its discovery in the 80s, the cannabinoid system has been extensively studied and described as being a relevant modulator of numerous physiological and pathological functions. This system comprises endogenous cannabinoids (endocannabinoids, EC), their receptors (CB), and the enzymes responsible for their synthesis and degradation. EC have a multifaceted role: in physiological conditions as psychoactive, analgesic, antiemetic, anti-inflammatory, vasorelaxant, orexigenic [19,20,21,22] and in a myriad of pathological states such as neurodegenerative disorders [23,24,25], myocardial infarction [25], liver fibrosis [26], and cancer [27,28]. Therefore, pharmacologic intervention of the EC system is a promising strategy for the management of many diseases. In fact, a substantial number of pharmacological agents that interfere with this system has been developed to date. Herein, we describe the most recent findings on the EC system, focusing on the pathogenesis of liver disease.

### 2.1. Endocannabinoids

EC are a class of arachidonic acid (AA) derivatives that interact with CB and were originally described as targets of Δ9-Tetrahydrocannabinol (THC), the main psychoactive constituent of *Cannabis sativa* [29]. EC share a common backbone structure resulting from their synthesis from membrane phospholipid precursors that contain AA and are conjugated either with ethanolamine or glycerol [22,26,30]. They are synthetized on demand, often in response to increased intracellular calcium concentrations [31]. EC amount is tightly regulated by changes in its catabolism rather by their synthesis. Figure 1 depicts the chemical structure of the main EC. Other less-characterized CB-interacting peptides and a series of AA derivatives that generate endocannabinoid-like effects such as N-palmitoylethanolamine (PEA) and Oleoylethanolamine (OEA) have also been described [32]. Most investigations focus on the first discovered and best studied EC: N-arachidonoylethanolamine (anandamide, AEA) and 2-arachidonoylglycerol (2-AG) [33,34].

AEA is a *N*-(polyunsaturated fatty acyl) generated from *N*-arachidonoyl phosphatidylethanolamine (NAPE) through multiple different pathways: cleavage by phospholipase D (PLD), sequential deacylation of NAPE by α,β-hydrolase followed by the cleavage of glycerophosphate, and phospholipase C-mediated hydrolysis of NAPE, which is then dephosphorylated by phosphatases [35]. It is a partial agonist of CB_1_ and CB_2_, presenting a lower intrinsic activity for the latter [29]. Fatty acid amino hydrolase (FAAH) is the main enzyme responsible for AEA degradation. However, it can also be catabolized via oxidation by cyclooxygenase-2 (COX-2) and by N-acylethanolamine-hydrolysing acid amidase (NAAA) [31]. 2-AG is an ester formed from AA-containing phospholipids and glycerol [31] via three major pathways: sequential activation of a phospholipase Cβ and a diacylglycerol lipase, sequential action of phospholipase A1 and a lyso phospholipase C, and by dephosphorylation of arachidonoyl- lysophosphatidic acid [36,37,38]. 2-AG is a full agonist of both CB receptors with moderate-to-low affinity [32]. Monoacylglycerol lipase (MAGL) is its principal degradation enzyme. 2-AG is also degraded by alpha/beta domain hydrolases 6 and 12 (ABHD6 and 12), oxidized by COX-2 and hydrolysed under some conditions by FAAH [31]. Although AEA and 2-AG are involved in similar processes such as the control of redox homeostasis and display anti-inflammatory effects, both agents are implicated in a myriad of different activities. For instance, AEA participates in cell cycle regulation and apoptosis, whereas 2-AG is important in synaptic signalling.

### 2.2. EC Canonical Receptors

EC mediate their cellular effects through two canonical CB: CB_1_ and CB_2_, numbered in the order of their discovery [20]. Both CB are seven transmembrane class A metabotropic G-protein-coupled receptors (GPCRs) but differ in amino acid sequence (48% homology in humans), tissue distribution, and signalling mechanisms [39]. Some EC actions may be mediated by other non-CB receptors including: G protein-coupled receptors (GPR3, GPR6, GPR12, GPR18, GPR55, and GPR119), transient receptor potential channels (TRPV1, TRPV2, TRPA1, TRPM8), ligand-gated ion channels, and nuclear receptors (for example, the peroxisome proliferator-activated receptor) [24,26,30,32,35,39,40,41]. Indeed, non-receptor targets such as cholesterol and cyclooxygenase-2 (COX-2) have been identified as interacting with them as well [40,41,42,43].

The CNR1 gene encodes a 472 amino-acid protein corresponding to CB_1_ in humans [32]. One canonical and two additional isoforms result from alternative splicing [44,45]. CB_1_ is the most abundant EC receptor and is exclusively responsible of the psychoactive effects of cannabinoids [22]. Its highest expression is found in the nervous system [31,39]. CB_1_ is also found to a lesser extent in vessels and peripheral tissues: skeletal muscle, spleen, tonsils, adrenal gland, bone marrow, liver, heart, lung, prostate, kidney, pancreatic islet, testis, and female reproductive tissues [32,39]. Mice deficient in CB_1_ have reduced progeny [46,47,48], show hypoactivity, hypoalgesia, enhanced spatial working memory, impaired contextual fear memory [48,49], and decreased insulin and leptin plasma levels [50].

CB_2_, encoded by the CNR2 gene, is composed of 360 amino acids in humans [32]. Two isoforms have been identified. CB_2_ is primarily expressed in the immune system: B cells, natural killer cells, spleen, bone marrow, tonsils, and pancreatic mast cells [51,52]. Functionally relevant expression has also been found in brain, myocardium, gut, endothelium, vascular smooth muscle and Kupffer cells, pancreas, bone, and reproductive organs [53]. CB_2_ knock-out mice display increased neuropathic pain, impaired formation of numerous immune cell populations such as splenic memory CD4^+^, and exacerbated inflammation as a result of enhanced monocyte and neutrophil recruitment [54,55,56].

CB can exist in dimers and complexes of higher magnitude [57], but the physiological relevance of dimerization has not yet been fully established. However, the expression of some heterodimers has been associated with different pathologies, for instance, cancer [58]. This suggests that EC can interact with multiple endogenous systems adding a higher level of complexity to the understanding of EC molecular mechanisms. Nevertheless, more data are needed to identify their specific biological functions.

### 2.3. Cell Signalling Pathways Activated by CB_1_ and CB_2_

Both CB are GPCRs that couple similar transduction systems of heterotrimeric G proteins [59]. Intracellular signalling is complex because it includes both G protein dependent and independent pathways, as well as ceramide signalling [53]. Figure 2 illustrates the main molecular pathways. Activation of CB decreases the intracellular accumulation of cyclic adenosine monophosphate (cAMP). This results in the inhibition of the adenylyl cyclase activity and a reduction in intracellular Ca^2+^, which eventually leads to the inhibition of protein kinase A and the dephosphorylation of K^+^ channel type A [60,61], a critical process that modulates the responses to ionotropic neurotransmitters in neurons. Moreover, CB_1_ negatively binds certain Ca^2+^ voltage dependent channels, induces G-protein-coupled modulating K^+^ channels (GIRK), and subsequently inhibits neurotransmitter release independently of cAMP [55]. CB also activate mitogen-activated protein kinase (MAPK) pathways, including extracellular signal-regulated kinase 1 or 2 (ERK1/2), p38, and c-Jun N-terminal kinase (JNK), which are important for orchestrating cellular functions such as cell growth, cellular transformation, and apoptosis [56]. It has been proposed that two signal transduction pathways activate MAPK in response to CB activation: one involves the activation of phosphatidylinositide-3-kinase (PI3K), and the second, sphingomyelin, hydrolysis and release of ceramide [57]. PI3K has also been reported to induce AKT-mediated cell survival by inhibiting apoptosis [56]. De novo ceramide synthesis promoted by CB is known to directly trigger apoptosis in a G protein-independent manner [62]. CB is also associated with β-arrestin, a key mediator of GPCR desensitization [32]. β-arrestin binds to the phosphorylated receptor and initiates its internalization process, during which it can mediate different signalling pathways [32].

## 3. The Endocannabinoid System (ECS) in Health and Disease

The ECS is widely distributed in the body and is involved in a wide array of physiological functions including pain, sleep, arousal, body temperature, feeding, and emotional regulation [63]. Disequilibrium of the ECS results in a myriad of pathological states. In the nervous system, CB_1_ activation is anti-emetic, enhances eating, causes changes in short-term and working memory, inhibits the release of neurotransmitters involved in anxiety and depression, attenuates nociception, and is protective against excitotoxicity [64,65,66]. In neurodegenerative diseases and psychiatric disorders, such as Alzheimer’s disease and schizophrenia, CB_1_ expression is reduced, whereas CB_2_ is induced in an attempt to halt inflammatory response [67,68]. Indeed, upon injury or inflammation, CB_2_ is highly induced to dampen inflammatory response [51]. CB_2_ activation decreases the production of reactive oxygen species (ROS) and the release of inflammatory mediators such as tumour necrosis factor α (TNF-α), and nitric oxide (NO) [68]. It activates immune cell apoptosis and suppresses immune cell activation, proliferation, and migration [69]. In the central nervous system, treatment with CB_1_ inverse agonists reduces the behavioural effects associated with drugs of abuse [63] and produces appetite suppression decreasing food intake, partially due to drug-induced food aversions such as nausea or vomiting [70]. In contrast, CB_2_ agonists block defecation, increase urination frequency, and produce analgesia [71]. Inhibition of EC catabolic enzymes such as FAAH seems also a good strategy to increase EC concentration and treat diseases such as anxiety, sleep disorders, and neuropathic pain [72].

In physiological conditions, the modulation of the ECS has a minor impact on the regulation of the cardiovascular system even though the ECS is widely distributed in the heart and blood vessels [17]. In contrast, ECS dysregulation results in disorders such as hypertension, myocardial infarction (MI), and chronic heart failure [62]. The activation of CB_2_ is cardioprotective, reducing cardiac fibrosis and infarcted areas during ischemia-reperfusion (I/R) and, at the same time, increasing the activation of cardiac progenitor cells and cardiomyocyte proliferation [73,74].

The ECS also plays a crucial role in lipid and glucose metabolism [63], displays orexigenic effects, and modulates peripheral energy metabolism [20]. In obese mice, stearoyl-CoA desaturase-1 generates monounsaturated fatty acids, which inhibit FAAH, resulting in increased AEA hepatic levels promoting excessive energy storage and insulin resistance [26]. In liver, CB_1_ stimulation increases de novo lipogenesis [20]. It has been suggested that CB_1_ has a key role in the development of diet-induced obesity and fatty liver disease [20]. Treatment with a CB_1_ inverse agonist SR141716 (rimonabant) improves glycaemic control, insulin resistance, and dyslipidemia [75]. However, it has no clinical use due to the significant psychiatric side effects: anxiety, depression, and suicidal ideation [63]. CB_2_ inverse agonists have also been proposed to modulate the metabolic syndrome, in particular triglycerides, insulin, adipose tissue, and arterial pressure [76].

The ECS also plays important roles in the gastrointestinal tract [77] and in oncology, not only as palliative but also as anti-emetic and anti-tumoural agents (due to their effect in controlling cellular proliferation, angiogenesis, and metastasis in experimental models of cancer [21,41,78]). However, there is controversy regarding their role as oncogenes or tumour suppressors [79,80].

## 4. Involvement of the ECS in the Pathogenesis of Liver Fibrosis

The role of ECS in fibrosis is complex and not yet completely understood since EC may mediate their effects in a CB receptor-dependent and independent manner. In early stages of liver disease, AEA and 2-AG levels are increased in rodents [81]. Furthermore, their serum and hepatic levels together with PEA and OEA are augmented in cirrhotic (CH) patients [82,83]. EC display a wide array of activities including anti-inflammatory, pro-apoptotic, and anti-proliferative [84]. Healthy hepatocytes have the highest level of expression and activity of FAAH in the body to tightly regulate AEA, PEA, and OEA levels in order to shield against cell death independently of CB receptors [85]. FAAH is decreased in murine models of BDL and CCl_4_, which promotes EC accumulation and, consequently, increased ROS and hepatocellular injury [86]. Activated HSC do not produce FAAH, and therefore, they are sensitive to EC-mediated apoptosis, a phenomenon which could be used as a promising therapeutic approach to attenuate the fibrogenic response [87]. Among all EC, 2-AG is known to be mainly metabolized by COX-2 to pro-apoptotic prostaglandin glycerol esters, which selectively induce HSC apoptosis [85,88]. OEA also ameliorates hepatic fibrosis by directly inhibiting tumour growth factor beta 1 (TGFβ1), signalling through the suppression of SMAD2/3 phosphorylation, a phenomenon inhibited in peroxisome proliferator-activated receptor alpha (PPAR-α) knock-out mice [89].

During acute and chronic liver injury, the expression of CB_1_ and CB_2_ is also augmented [90,91]. CB_1_ is strongly induced in hepatocytes, HSC, endothelial, inflammatory, and ductular proliferating cells, whereas CB_2_ is mainly induced in inflammatory cells and HSC [92]. A large number of studies have addressed the role of CB in liver pathogenesis elucidating that both exhibit opposite effects. CB_1_ activation promotes hepatic steatosis, inflammation, and fibrosis in non- and alcoholic fatty liver diseases [93,94]. CB_1_ stimulation also contributes to the progression of cirrhosis by triggering fibrogenesis. In contrast, genetic and pharmacological inactivation of CB_1_ with rimonabant enhances HSC apoptosis and decreases the proliferative response to platelet-derived growth factor (PDGF), reducing TGFβ1 levels and liver fibrosis [92]. Systemic administration of rimonabant also results in increased arterial pressure and peripheral resistance and a reduced mesenteric blood flow and portal pressure in CCl_4_-exposed rats, an effect not seen in healthy animals [95,96]. In line with these results, rimonabant reduces the incidence and accumulation of ascites and also improves sodium balance, delaying decompensation in CH rats [97]. Rimonabant has also been shown to reduce the matrix methaloproteinase (MMP) abundance and activity, the expression of the pro-fibrogenic factors endothelin 1 (ET-1) and TGF-β, and the synthesis of the pro-inflammatory factors TNF-α and monocyte chemoattractant protein-1 (MCP-1) in CCl_4_-treated cirrhotic rats [98]. In addition, CB_1_ antagonism with rimonabant protects against lipopolysaccharide (LPS)-enhanced liver injury by interfering with inflammatory response [99]. Circulating LPS, which is common in CH patients, has been reported to increase de novo AEA and 2-AG production in monocytes and platelets and activates CB_1_, suggesting a key role of LPS in triggering the EC synthesis in the fibrotic liver [100]. These EC are also highly synthesized in adventitial cells and detected at a lower concentration in endothelial cells from CH mesenteric vessels, although they can also be found at lesser amounts in healthy vessels [101]. Monocytes isolated from CH patients and then injected to control rats promote CB_1_-mediated hypotension [90]. CB_1_ is also involved in other complications associated with liver cirrhosis, including cardiomyopathy and encephalopathy [102], and at the same time, in the process of liver regeneration [103].

CB_2_ exerts anti-fibrogenic and anti-inflammatory effects [104] and protects against liver I/R injury [105]. CB_2_ knock-out mice exhibit enhanced fibrosis, inflammation, and steatosis compared to wild type animals [19]. These mice are more susceptible to hepatic insult, enhancing MMP-2 activity as a consequence of IL-6 down-regulation [92]. Activation of CB_2_ by the selective agonist JWH-133 reduces inflammation and fibrosis and increases arterial pressure and non-parenchymal cell apoptosis in CH rats [106]. JWH-133 administration in CCl_4_-treated mice reduces liver injury and accelerates the regenerative response [92]. JWH-133 also reduces liver fibrosis and inflammation by decreasing IL-17 production via STAT5-dependent signalling in BDL rats [107]. In agreement with these findings, long-term administration of the CB_2_ agonist AM-1241 significantly inhibits PDGF signalling, reduces HSC activation and hepatic fibrosis, and improves hemodynamic function in fibrotic rats [104].

Increased levels of endogenous AEA have also been found in the heart of cirrhotic rats, where it contributes to the reduced cardiac contractile function that leads to cirrhotic cardiomyopathy [90]. In the brain of mice, 2-AG has also been found to be elevated with thioacetamide-induced hepatic encephalopathy. Indeed, the administration of a CB_2_ selective agonist improves the cognitive function in these mice [108]. These findings point to ECS as a potential target for the treatment of hepatic encephalopathy secondary to decompensated cirrhosis.

All the investigations and evidence so far suggest that the ECS may be a crucial regulator in different liver diseases. Both CB_1_ blockade and CB_2_ receptor stimulation have been successful in preventing fibrosis progression in experimental animal models. Figure 3 illustrates the potential roles of CB_1_ antagonists and CB_2_ agonists in the intervention algorithm for the treatment of liver steatosis, fibrosis, cirrhosis, and its complications. CB_1_ antagonists additionally ameliorate systemic haemodynamics. However, since CB_1_ is highly expressed in the central nervous system, treatment with antagonists is often associated with undesirable central effects. Therefore, to date, treatment with CB_2_ agonists seems to be a more feasible approach to treat liver fibrosis and cirrhosis and its complications.

Further studies are required to elucidate the precise mechanism by which EC participate in the pathophysiology of liver disease and more long-term studies are needed to confirm the absence of central effects of selective CB_1_ antagonists. In addition, further efforts should be made in the generation of specific compounds able to modulate EC anabolism, transport, or catabolism since this could be a promising strategy to modulate the ECS activation tone. The future of EC therapeutics in fibrosis might be addressed to combine a CB_2_ agonist with another efficient anti-fibrotic agent to synergize activities. In line with this, one study has shown that the CB_2_ agonist AM-1241 and the apelin receptor antagonist F13A display similar mechanisms of control of HSC cell activity despite the clear differences between the CB_2_ and APJ signalling pathways [104]. The combination of different drugs with anti-fibrotic activity could be an interesting approach to improve pharmacological therapies in liver fibrosis.

## 5. Overview of the Apelin System

Apelin is a peptide that was first described in 1998 as the endogenous ligand for an orphan receptor called angiotensin-like-receptor 1 (AGTRL1, also known as APJ), a G-protein-coupled receptor [109]. During the last two decades, this receptor has been involved in an array of physiologic events, such as water homeostasis [110], regulation of cardiovascular tone [111], and cardiac contractility [112] as well as in chronic liver disease [18]. Apelin and its receptor are expressed in the central nervous system and in peripheral tissues, especially in endothelial cells as well as in HSC, leukocytes, enterocytes, adipocytes, and cardiomyocytes [113,114,115,116,117,118]. Since the discovery of the apelin/APJ interaction, numerous investigations have emerged highlighting new roles for the apelin system in the regulation of different homeostatic processes or involving apelin/APJ in disease. Here, we briefly review the components of the apelin/APJ system as well as the molecular mechanisms involved in the diverse cellular effects observed so far before focusing on the relationship between the apelin system and the pathogenesis of liver fibrosis.

### 5.1. Apelin: Peptide Isoforms

The human apelin gene (APLN) encodes a 77 amino acid prepropeptide that can be cleaved into different fragments [119]. Endopeptidases cleave the apelin prepropeptide into a 55 amino acid molecule, which in turn is fragmented by the angiotensin-converting enzyme 2 (ACE2) generating the following bioactive isoforms: apelin 36, apelin 17, apelin 13, and apelin 12 (Figure 4). This family of apelin peptides displays a wide array of biological functions in mammals including the neuroendocrine, cardiovascular, and immune systems [120] that can be explained by their asymmetric tissue distribution, activity and receptor binding affinity. Indeed, the shortest apelin isoforms (apelin 12 and 13) demonstrate the highest agonist activity on APJ [117,121]. Apelin 13 can act via autocrine, paracrine, endocrine, and exocrine signalling and is the main agent responsible for the APJ stimulation and downstream biological activities of mature apelin [122]. Several investigations have reported that the transcriptional regulation of apelin expression is modulated by some conditions, hormones, and inflammatory factors. Induction of apelin expression has been associated with hypoxia [123,124,125], activity of hormones [118,126,127,128,129,130,131,132], and inflammatory factors [115,133,134] in different cell types and tissues. Apelin knock-out mice show reduced retinal vascularization and ocular development, denoting its major role in angiogenesis [135], which is independent of VEGF and FGF receptors [136]. Moreover, a lack of apelin in knock-out mice increases angiotensin II-induced dysfunction and pathological remodelling [137].

### 5.2. APJ, the Apelin Receptor

APJ is a G-protein-coupled receptor composed of seven transmembrane domains with a 31% of homology with the receptor 1 of angiotensin [138]. For this reason, it was initially called angiotensin-like-receptor 1. However, it has been demonstrated that angiotensin II (AII) has no affinity for APJ. In fact, the activation of vascular APJ displays a counterregulatory effect against the activation of the AII receptor [139]. In normal conditions, lung, spleen, and mammary glands show the highest APJ expression [117,121]. APJ is coupled to an activator G protein (Gq) and to an inhibitor G protein (Gi), the activation of which results in an array of physiological effects such as the regulation of hydrosaline equilibrium, vascular tone, cardiac formation and contractility, angiogenesis, and HSC activation [111,124,132,140,141]. The APJ gene sequence has no introns. APJ gene expression is modulated by hypoxia [124,142], insulin [143], stress, and glucocorticoids [144]. However, the molecular mechanisms of APJ transcriptional regulation have not been extensively characterised to date. Mice deficient in APJ do not show any relevant phenotype apart from a higher vasoconstrictor response to AII [110]. During the last few years, another peptide called Apela/ELABELA has been described as an APJ activator, displaying similar effects to apelin on the cardiovascular system [145,146,147]. The possible interactions of this new APJ agonist and the clinical implications of Apela in the apelin system are still under investigation.

### 5.3. Cell Signalling Pathways Activated by the APJ Receptor

On one hand, APJ activation leads to the activation of phospholipase Cβ, the phosphatidylinositol-3-kinase (PI3K)/Akt pathways and the Na/H exchanger type 1 and, on the other hand, inhibits adenylyl cyclase and subsequent cyclic adenosine monophosphate production [111,148,149,150,151]. Phospholipase Cβ triggers protein kinase C (PKC) and downstream Ras/Raf/MEK/Erk, which together with Akt via mTOR are involved in the activation of P70S6K [152] and the endothelial NO synthase, promoting the release of NO, vascular dilatation, and cell proliferation [149]. The activation of the Na/H exchanger type 1 via PKC in cardiomyocytes is responsible for the dose-dependent increase in in vivo and in vitro myocardial contractility [153]. Moreover, some studies associate APJ activation with inflammation [113,115,134]. Actually, APJ induces the expression of vascular cell adhesion molecule-1 (VCAM-1), MCP-1 and intercellular adhesion molecule-1 (ICAM-1) via the NF-κB and the Jnk signalling pathway [154] (Figure 5).

## 6. The Apelin System in Health and Disease

The apelin system displays major physiological roles in vascular and lymphatic development [155,156], in neurology [157], and in the digestive system [158]. The activation of the apelin system has demonstrated many beneficial effects in cardiovascular, kidney, skin, and metabolic diseases [128,142,150,159,160,161,162,163,164]. Apelin has aroused a special interest in the field of cardiology since it is one of the most powerful dose-dependent positive inotropic agents known to date, as demonstrated in perfused hearts [153]. Moreover, apelin is a very-well known vasodilator involved in the stimulation of NO vascular release [165]. Indeed, APJ agonism shows sustained and preserved local vascular and systemic hemodynamic responses in patients with stable symptomatic chronic heart failure and standard medical therapy [166]. However, there are some controversial data on the precise role of the apelin system in the pathogenesis of human heart failure. Some reports suggest that the apelin/APJ system is down-regulated in heart failure and upregulated in left ventricular remodelling [167,168]. This might point to a systemic compensatory effect to recover cardiac contractility. Indeed, acute administration of apelin restores cardiovascular functions in chronic heart failure [169]. However, the potential utility of apelin in cardiovascular disease needs further investigation.

The use of apelin as a biomarker in human heart failure has been challenging. Some reports point out that apelin does not reliably predict acute heart failure in patients presenting dyspnoea, and it is not a prognostic marker in those with confirmed heart failure or with chronic heart failure secondary to idiopathic dilated cardiomyopathy [170,171]. In contrast, plasma apelin concentrations add prognostic value in conjunction with brain natriuretic peptide (BNP) to the risk of mortality at 6 months in patients with ST-segment elevation myocardial infarction [172].

## 7. Involvement of the Apelin System in the Pathogenesis of Liver Fibrosis

The apelin system has certain therapeutic abilities, but it may also play a role in disease depending on the cellular and molecular milieu. A clear example of this is the role of apelin in tissue fibrosis. Apelin displays anti-fibrotic actions counteracting AII in models of cardiac fibrosis [173,174] and renal fibrosis [175,176]. In contrast, the scenario and role of the apelin system are completely different in liver disease. Apelin is a hepatic pro-fibrotic agent, in part by mediating some of the fibrogenic effects triggered by AII and ET-1 in the activation of HSC occurring in liver fibrosis [118]. These outcomes highlight the intriguing difference between myofibroblastic-cell types in liver compared to myofibroblasts in heart, kidney, or skin in which apelin acts by decreasing myofibroblast accumulation and activity [137,176,177,178]. In vitro, apelin has also demonstrated a significant potential to promote liver fibrosis. It acts directly on LX-2 cells (a cell line used as a reliable in vitro model of HSC) through Erk signalling [179], stimulating cell survival and the synthesis of PDGF-β receptor and collagen-I in these cells [118]. In turn, PDGF-β and LPS can stimulate the expression of APJ, expanding and perpetuating HSC activation [180]. Therefore, apelin may, in theory, induce HSC to a pro-fibrogenic profile and prolong its stimulation autocrinally during all stages of liver fibrosis in chronic liver disease (Figure 6). In fact, some data have revealed that the inhibition of APJ using F13A (an APJ antagonist) prevents fibrosis progression in rats under a non-discontinued fibrosis induction program using CCl_4_ [104].

Several investigations have uncovered the close and integrated relationship between pathological angiogenesis and fibrosis [12,181,182,183,184,185,186]. As mentioned above, apelin is a powerful angiogenic agent through the activation of endothelial APJ and different downstream signalling pathways. Two reports have associated the inhibition of APJ with a reduction in angiogenesis and with a concomitant drop in fibrosis in CH and fibrotic rats [18,104]. Although a direct relationship was not established between the two phenomena, there are evidence pointing out that APJ activation by apelin stimulates the expression of a well-known pro-angiogenic factor, angiopoietin-1 (from the angiopoietin family involved in pathological angiogenesis in chronic liver disease) [185,187] in LX-2 cells [118]. This suggests that fibrogenic cells such as HSC may participate in hepatic angiogenesis by secreting angiogenic factors such as angiopoietin 1. Indeed, the inhibition of HSC-secreted angiopietin-1 has shown to drastically reduce pathological angiogenesis and liver fibrosis induced in mice by either CCl_4_ or BDL [187].

Aside from HSC, hepatocytes have also been related to contribute to both phenomena, pathological angiogenesis and liver fibrosis, by releasing pro-angiogenic and pro-fibrogenic factors [186]. There is a growing belief that local hypoxia is the link interconnecting both pathological angiogenesis and liver fibrogenesis, orchestrating the harmonic and coordinated activation of HSC, hepatocytes and other hepatic cells [188]. Detection of hypoxic areas is a common trait at any stage of chronic liver disease, expanding progressively from early injury to the development of cirrhosis [189]., Through the action of hypoxia-inducible factors (HIF), hepatic hypoxia up-regulates the expression of a wide array of growth factors and mediators of liver repair and angiogenesis [189]. However, pathological angiogenesis can be inefficient due to the immaturity and permeability of vascular endothelial growth factor (VEGF)-induced new vessels [190] and, consequently, the liver may be unable to reduce hypoxia. Hypoxia up-regulates in vitro the expression of APJ in LX-2 (HSC) and in HepG2 (hepatocytes) [124]. Interestingly, the pro-inflammatory and pro-fibrogenic agents TNF-α and AII also induce the expression of APJ in HepG2 cells [124]. APJ activation in HepG2 cells triggers the expression of VEGF-A and PDGF-β, factors that in turn may promote angiogenesis and activation of HSC and, consequently, liver fibrosis [15,191]. According to these data, hypoxia, inflammation and pro-fibrogenic factors up-regulate APJ in HSC and hepatocytes, which can release different pro-angiogenic and pro-fibrogenic factors such as apelin (from activated HSC) together to perpetuate fibrosis while injury and these stimuli remain. In vivo, APJ is upregulated preferentially in hepatocytes and HSC, while apelin levels are increased and localized in HSC in cirrhotic rats and in patients with liver cirrhosis caused by hepatitis C virus or ethanol [18,124]. High apelin levels and liver damage have also been observed in human non-alcoholic fatty liver disease [192]. A contemporary clinical investigation revealed that circulating apelin levels are associated with histological and hemodynamic features of chronic liver disease [193], but more clinical studies are needed to confirm the major relevance of apelin system in human liver fibrosis.

One study suggested that apelin may play a different role in liver fibrosis by being an initiator of hepatic injury instead of merely a HSC activator following exogenous injury [194]. In this study, the authors described that the apelin system may stimulate Fas-induced liver injury via the phosphorylation of Jnk in mice intraperitoneally injected with an agonistic anti-Fas antibody. Similar results have been obtained when acute liver injury was promoted by hepatectomy. Blockade of the apelin system resulted in mouse liver regeneration via activation of Kupffer cells and by increasing TNF-α and IL-6 levels in hepatectomized mice [195]. These results suggest that the apelin system may interfere with hepatocyte proliferation after partial hepatectomy in mice. However, a recent study has shown that the administration of a long-acting apelin fusion protein resulted in attenuated hepatocyte damage, diminished apoptosis and ROS production in a mouse model of LPS-induced liver injury [196]. Altogether, these findings suggest that the apelin system may be involved in the processes of hepatic injury and regeneration, but these specific aspects need further investigation.

## 8. Conclusions

The EC and apelin systems are two of the multiple cell-signalling pathways involved in the pathogenesis of liver fibrosis. Both systems play a major role in the pathophysiological mechanisms underlying the control of HSC activity, involving different receptors and molecules, but with a common significant impact in the development of liver fibrosis.

The ECS is upregulated in liver disease and has been associated with hepatic steatosis, regeneration, fibrosis, and cirrhosis. In liver fibrosis, the cannabinoid receptors CB_1_ and CB_2_ exhibit opposite roles: CB_1_ activation accentuates hepatic fibrosis progression whereas CB_2_ displays anti-fibrogenic and anti-inflammatory activities. CB_1_ also contributes significantly to cirrhosis complications including portal hypertension, splanchnic vasodilation, cardiomyopathy, and encephalopathy. More specific CB_1_ antagonists and CB_2_ agonists need to be developed as current CB_1_ antagonists are able to cross the blood–brain barrier causing psychotic side effects. Compounds targeting EC synthesis, degradation, and cellular transport pathways could also be a valuable approach to modulate the ECS. Several efforts have been devoted to understanding the specific pathways regulated by this system but there is still a long way to go in the development of drugs targeting the ECS.

Recent studies have reported multiple roles for the apelin/APJ system in liver disease, including acute liver injury, regeneration, fibrosis progression, and cirrhosis. Apelin/APJ has unique functions as a regulator of cell proliferation, apoptosis, pro-inflammatory activity, and revascularization. Apelin/APJ gene expression is temporally increased during liver cirrhotic development and is decreased in stabilized liver fibrosis. The validation of using apelin/APJ as a biomarker in different liver diseases would also be a crucial step toward its clinical use. Further experimental or clinical findings will help to determine the potential of therapeutic strategies targeting the apelin/APJ system for the treatment of liver disease.

Future investigations to further define the mechanisms by which the EC and apelin systems contribute to modulate liver fibrosis will enhance our understanding of their cellular and molecular mechanisms and possible therapeutic targets. This understanding will eventually help in the development of novel therapeutic strategies and drug candidates for treating liver fibrosis in patients with chronic liver disease.

## Figures and Tables

**Figure 1 cells-08-01311-f001:**
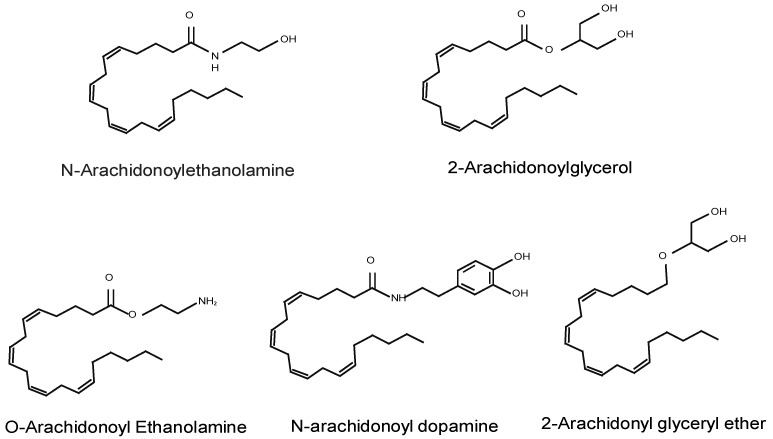
Chemical structures of endogenous cannabinoids: *N*-arachidonoylethanolamine (AEA, anandamide), 2-arachidonoylglycerol (2-AG), *O*-arachidonoyl-ethanolamine (*O*-AEA, virodhamine), *N*-arachidonoyl dopamine (NADA), and 2-arachidonyl-glyceryl ether (2-AGE, noladin ether).

**Figure 2 cells-08-01311-f002:**
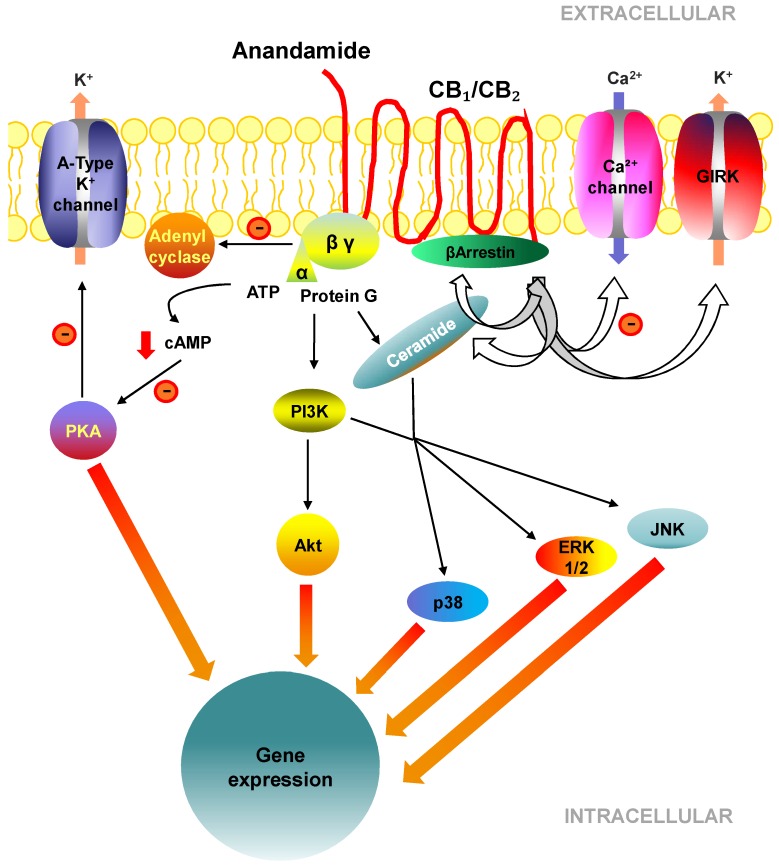
Molecular and cell signalling pathways of cannabinoid receptors.

**Figure 3 cells-08-01311-f003:**
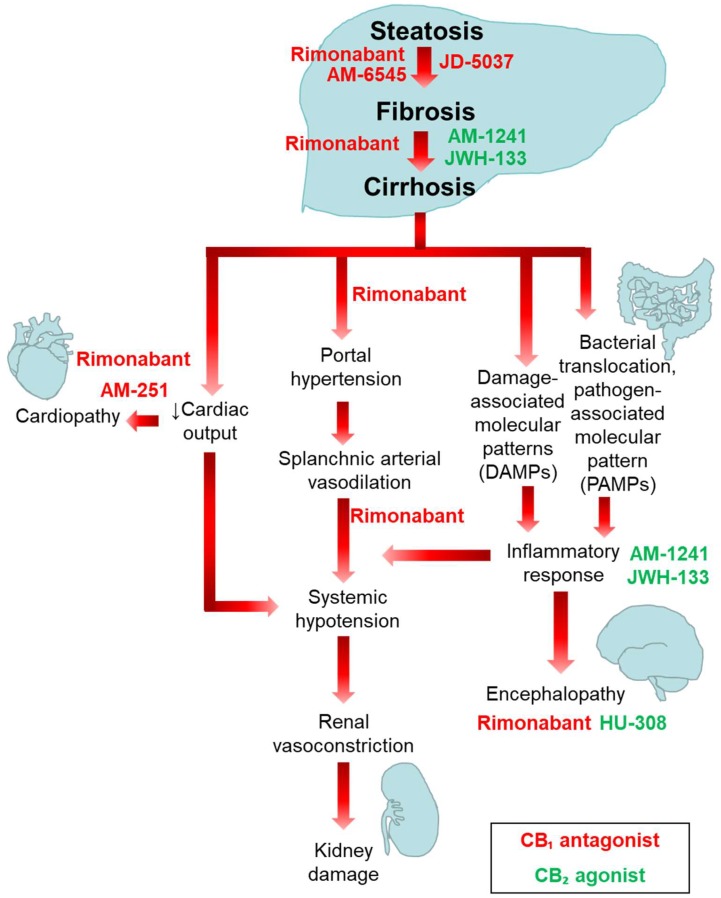
CB_1_ antagonists and CB_2_ agonists as potential drugs for the treatment of liver cirrhosis.

**Figure 4 cells-08-01311-f004:**
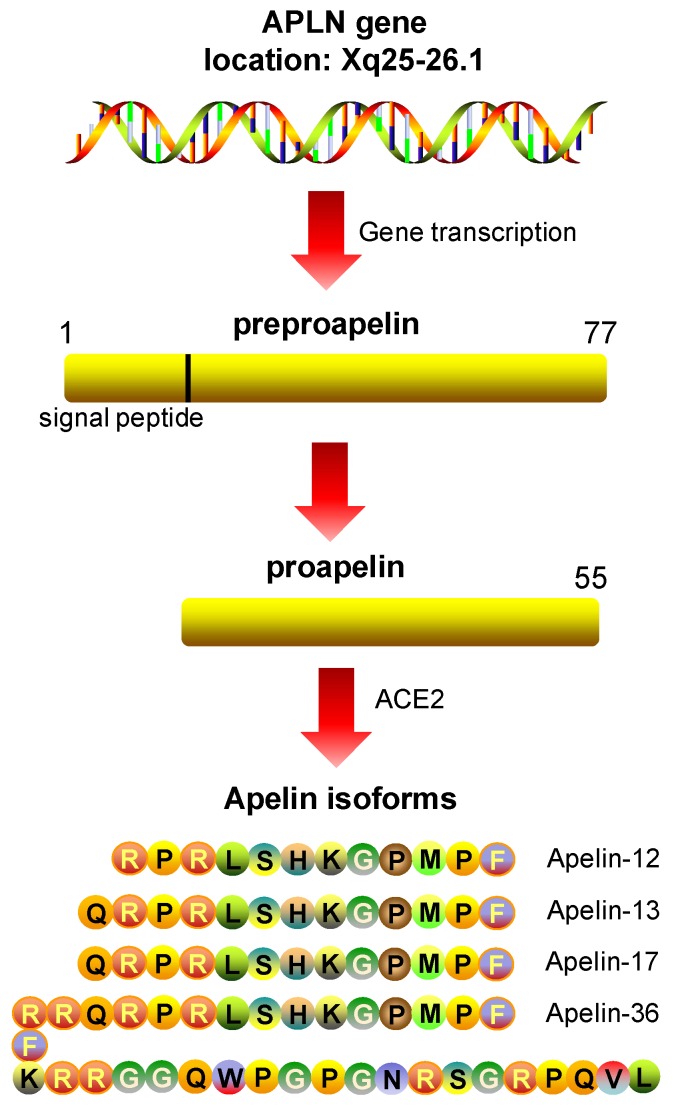
Apelin peptide isoforms.

**Figure 5 cells-08-01311-f005:**
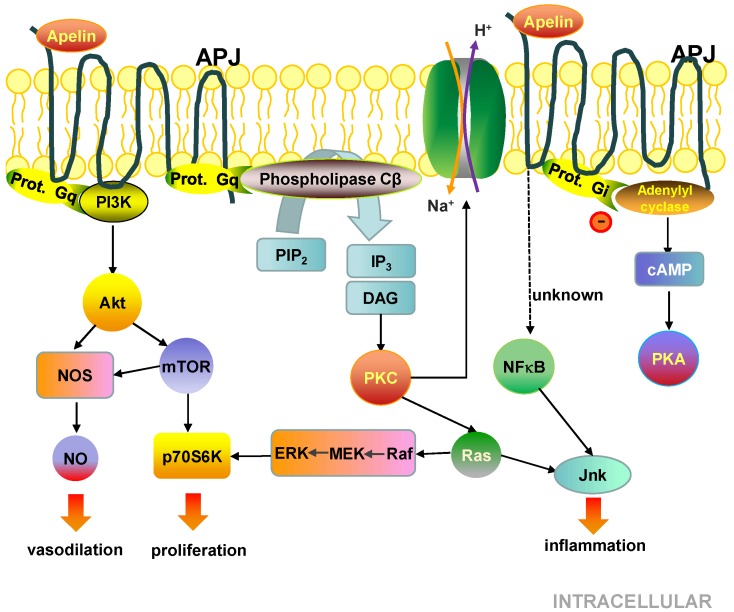
Molecular and cell signalling pathways of APJ.

**Figure 6 cells-08-01311-f006:**
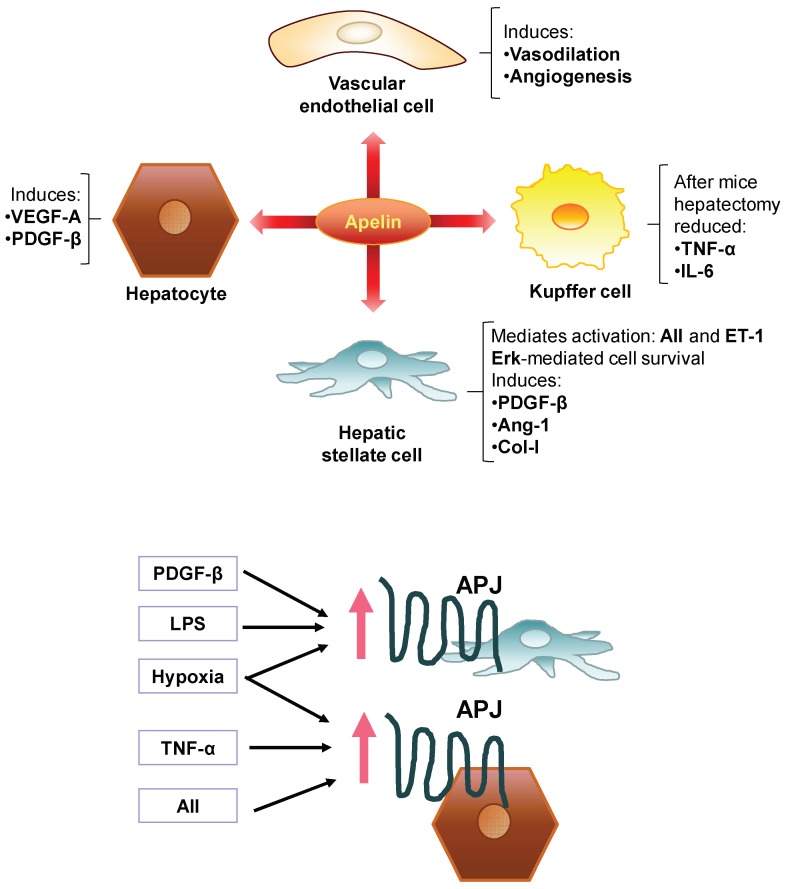
Overview of the roles of the apelin system in liver fibrosis.

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
