# Peer review of "Roles of the Hepatic Endocannabinoid and Apelin Systems in the Pathogenesis of Liver Fibrosis"

_cells, 2019, doi:10.3390/cells8111311_

Round 1

Reviewer 1 Report

This is a very interesting and timely review paper that covers all important aspects of the role of endocannabinoid system in liver fibrosis pathogenesis.

This reviewer has the following minor suggestions to be considered by the authors:

Please, specify why, out of of numerous contributors/cell signalling pathways in pathogenesis of hepatic fibrosis, the endocannabinoid and apelin systems were chosen.   While both endocannabinoid and apelin systems are comprehensively described, less attention was devoted to general mechanisms of hepatic fibrosis. This reviewer believes that this manuscript will significantly benefit by adding a section on pathogenesis of hepatic fibrosis as well fibrosis-preceding pathological states, such as (non)-alcoholic fatty liver disease. Lines 191-193 - please, elaborate more on the data available from the CCl4-treated rats and  effects associated with administration of selective CB1 antagonists: are these effects considered positive or negative?

Reviewer 2 Report

The submitted review article concerns the role of endocannabinoid and apelin systems in liver fibrosis and fully fits the aim of the special issue.

In general the manuscript is clear and well written, but its main weakness point is the lack of integration/relationship between endocannabinoid and apelin system. How do the authors justify the decision to compare endocannabinoid and apelin systems? What is their relationship, if any?

In fact, it seems like to read two different review articles with a common introduction about liver fibrosis and a summary of the major concerns in the conclusion section.

Additional points:

The section on endocannabinoids has been mainly focused on CB1/CB2 than on the system as a whole. Endocannabinoid activity strongly depends on endocannabinoid tone which is deeply modulated by FAAH hydrolase. In this respect, FAAH is an additional possible therapeutic/diagnostic biomarker. In spite literature in the field, this point has never been discussed in the manuscript. Thus the article is focused on the activity of cannabinoid receptor in the liver and not on hepatic endocannabinoid system.  The paragraphs focused on endocannabinod/apelin system in health and disease have a general title but are mainly focused on cardiovascular disorders/angiogenesis. The use of abbreviations is not consistent all over the main text and figures (as for instance see anandamide AEA or rimonabant SR141716). Not all abbreviations have been defined at the first appearance in the main text. 88 “EC receptors (Structure and location)”: This paragraph lists the canonical and not canonical EC receptors, focuses on CB1/CB2 and summarizes their tissue expression, thus (structure and location) is not suitable. 90-91 “Both CB are seven transmembrane class A metabotropic G-protein-coupled receptors (GPCRs) but differ in the amino acid sequence (48% homology)”: specify the specie in which such an homology has been detected 98-99 “The CNR1 gene encodes for CB1 in humans, and consists of 472 amino acids”: this sentence requires changes in that the gene does not consist of 472 amino acids 101-103: among peripheral tissues expressing CB1, the authors forget female reproductive tissues The authors generally use “inverse agonists”; due to the large number of synthetic agonists/antagonists-all exhibiting specific properties with respect to cannabinoid receptors-   they have to specify what agonist  are talking about. Figure 3: see the previous comment

Round 2

Reviewer 2 Report

The authors have carefully revised their manuscript and addressed most queries. The revised version of the manuscript has been improved and is better organized. The choice of putting together endocannabinoids and apelin remains- in my opinion-a bit forced, but focusing on liver fibrosis the manuscript surely deserves attention.

Lastly, the rebuttal letter is quite confusing in that suggested changes not always corresponds to the changes marked by the authors. Just to make an example. In the rebuttal letter, the authors declare: “We agree with the reviewer that this differentiation was not clear enough, and therefore we have clarified this point in the new version of the manuscript (lines 294 - 295 and 471 - 474)”, but 294 – 295 concern the description of apelin and 471 – 474 are within reference list.